# Evaluation of Older People Digital Images: Representations from a Land, Gender and Anti-ageist Perspective

Georgiana Livia Cruceanu [1,*], Susana Clemente-Belmonte [1], Rocío Herrero-Sanz [2], Alba Ayala [3,4], Vanessa Zorrilla-Muñoz [4,5], María Silveria Agulló-Tomás [2,4], Catalina Martínez-Miguelez [2] and Gloria Fernández-Mayoralas [4,6]

[1] Faculty of Political and Social Sciences, Complutense University of Madrid, 28223 Madrid, Spain
[2] Department of Social Analysis, University Carlos III of Madrid, 28903 Madrid, Spain
[3] Statistics Department, University Carlos III of Madrid, 28903 Madrid, Spain
[4] University Institute on Gender Studies, University Carlos III of Madrid, 28903 Madrid, Spain
[5] Fundación Pilares para la Autonomía Personal, 28003 Madrid, Spain
[6] Spanish National Research Council (CSIC), Institute of Economics, Geography and Demography (IEGD), 28037 Madrid, Spain
* Correspondence: georgicr@ucm.es

**Abstract:** There are numerous sociological and psychosocial studies, both classic and current, that have analysed the images and representations of older people and aging. If gender, intersectional and land perspectives are added, the literature consulted is only a few years old, particularly in Spanish. In addition, research based on fieldwork from virtual image banks is still scarce and recent. The objective of this paper is to evaluate the images from some free access image banks (like Freepik, Canva, Pixabay, or Storyblocks) of older people from a gender, intersectional and socio-spatial and land perspective. Methods: 150 images have been analysed following different selected criteria: 22 variables related to gender, activity, socio-spatial environment, natural space and land, among others, briefly describe the main methods or treatments applied. The key results show a stereotyped and barely diverse image of old age and aging around positive representations, with a notable absence of images related to loneliness as opposed to the presence of social relationships. A feminization has also been observed in the representations, with an imbalance in the activities that are carried out (care in the case of women and leisure in the case of men) and in the visible space (indoor among women and outdoor among men). Older people are still identified with a rural, traditional, and more defined territory and not with more diverse and ecological spaces, which are more frequently attributed to younger profiles. This evaluation contributes to linking this necessary connection of current issues and challenges to ageism, sexism and other exclusions derived from territory and socio-spatial aspects. However, more research is still needed, and, in fact, a second phase of the fieldwork is underway to broaden the sample and to expand further evaluations of images.

**Keywords:** land; environmental issues; older people; gender; images; evaluation



## 1. Introduction

Images, which are central to audio-visual culture, arise special interest as qualitative and quantitative material for analysis and social research. However, these digital artifacts [1] give rise to theoretical debates concerning the subjectivity/objectivity of the images, debates from which the mirror metaphor emerges as a critique [2]. From this approach, images are not characterized by reflecting reality itself, i.e., they are not only a window from which one can view the world, but rather a reflection of stereotypes, prejudices, discourses, and cultural codes of the subject that creates the image [2]. In turn, images construct a collective identity of older people and shape a social imaginary about old age [3,4]. Indeed, the analysis of images (and their meanings) provides a glimpse of

the social construction of old age [5] or old ages, in plural, although this mosaic of social representations is still not diverse [6].

Unlike social networks, where users are the ones who represent themselves in the virtual space and maintain or create new identities [7], image banks are portals of stock photographs, illustrations, videos, vectors, or icons, among other resources created by professionals, and are used in multiple areas, thus allowing the democratization of the use of images [8]. In the context of the current social and digital transformations, the consumption and production of still images acquires a certain complexity [9] as well as a privileged position by being at the centre of people's lives [10]. Images offer "[individuals and collectives] the possibility of feeling represented, self-defining, capable of interpreting or understanding reality" [10]. In other words, its study "contributes to the understanding of the social world by revealing and updating aspects that are not conscious" [11], such as sexism and ageism, which may go unnoticed, but which, together with racism, are the main forms of discrimination [12].

The perception or social imaginary of old age in Western culture has been constructed negatively, "we associate old age mainly to the decline of all aspects of personal and social life" [13]. Moreover, Simone de Beauvoir classified it as an immutable social construct [14], and it still represents a problem of social and environmental inequalities [15].

Since the production and reproduction of images, values and stereotypes has a great impact on the individual and social levels, there is a need to investigate and evaluate the images that represent old age and older people. The purpose of this article is to answer a series of research questions about the images associated with old age, the way in which older persons are represented and where they are located. In answering these questions, we also consider related aspects such as dependency, the leisure and free time activities in which they participate, the social relationships they maintain, or their emotional state, along with the application of a gender, anti-ageist, intersectional and inclusive perspective.

*Background*

Studies on the relationship between images, aging, gender and the environment are still scarce but sufficient to show the existence of both an academic and a social interest on the matter. It should be noted that both classical and recent authors, especially from Sociology, Social Psychology or Communication, have dealt with the importance that images (stereotypes, labels, social representations...) have for the identity and better self-perception of the population. However, the references that allude to the images of older people from a gender, socio-spatial and intersectional prism are scarce.

Image banks comprise primary information, the images per se, and secondary information, which includes image description and further information [16], although the latter is not always available. When this happens, there are other ways to obtain information, like with descriptive labels, titles that briefly indicate what the image is intended to represent or with the similar images proposed by the image bank. Thanks to metadata, understood by Mevillet [17] as "the structured set of data describing a resource, book, image, article, video, audio document, etc.", it will be possible to make use of images as an analytical and interpretative tool [4].

The reasons for accessing image banks are diverse; for example, to incorporate an image, to contextualize a newspaper article or to use for an Instagram post. It is necessary to point out that the access and use of these images is not cheap, so not everyone has the economic capacity to acquire them: we are talking about paid image banks such as iStock, Dreamstime or Depositphotos. However, there is a wide range of open access and free-to-download image banks, either with or without the need to register on the platform.

Regarding image evaluation, previous studies have used this audio-visual methodology. One can find, for example, the study conducted by Anne-Vinciane Doucet in 2008 [17] in which she analysed the description of 30 images from environmental image banks. Other studies focus their attention on aspects such as copyright management systems [18], image banks and their characteristics [19] or on the type of images available, their conditions of

use and collaborative creation [20]. From another point of view, the evaluation of images has not taken place exclusively in digital spaces; authors such as May Narahara (1998) [21] or Angela M. Gooden and Mark A. Gooden (2001) [22], among other studies in this line, conducted evaluations of images found in textbooks. Others emphasize the image of older people in mass media or in movies [23].

Objective information on old age and aging can be found in statistical institutes. In Spain, the National Institute of Statistics (or Instituto Nacional de Estadística in Spanish), as of 1 January 2021, puts the population aged 65 and over at 9.38 million people, which makes up almost one-fifth of the total population [24]. To this, the phenomena of the feminization of old age (there are more women than men and their life expectancy is higher), of masculinization (overrepresentation of men) in some rural areas, and the aging of aging (increase in the number of people aged 80 and over) are added, which imply the presence of groups of people in more need of attention and care [25].

The most prevalent approaches to old age both in the academic debate and in the social discourse are associated with the most negative aspects that focus on the one hand, on the deterioration of functional capacities and, on the other, on the non-production of goods after retirement. These discourses can negatively affect younger generations in their perceptions and behaviours toward older persons [26], influence professional care for this group [27], and affect self-perceptions about aging itself, which, in turn, can impact health, performance, and longevity [28]. In a meta-analytic review, it was observed that the most negative attitudes occur towards older people and that women are doubly stereotyped for being female and older [29]. Indeed, in the case of women the prejudice is twofold since the sexist stereotype (where the reproductive role of women loses meaning) is added to this negative view [30].

In contrast to the traditional mainstream media discourse on old age as a decadent, unproductive and dependent stage, there is a more recent positive discourse based on the concept and framework for action to promote Active Aging, proposed by the World Health Organization at the Second World Assembly on Aging held in Madrid in 2002 [31–33]. Some authors warn of the risks that an idealized vision of old age may entail, thus becoming a potential source of frustration for older people when certain goals prove unattainable [34].

Sociological studies on images and women, or images and older persons, are numerous. However, research that considers age and gender intersectionality, that is, focused on the study of images and older women, is much scarcer. This assertion is based on the bibliographic approach to studies on images, aging, and gender.

Regarding spaces and the environment, the residential environment can be considered as the closest geographical space where the older people population maintains its social relations and develops its life [35], which is constituted, therefore, by the home and the neighbourhood. In this way, the state of housing, its ownership [36] and a better assessment of the closest residential environment will directly affect the well-being and quality of life of the older persons [35,37]. Furthermore, the characteristics of the physical environment, social and family relations, accessibility and safety are key to the promotion of leisure time activities that help encourage active aging [38].

Indeed, during the aging process, physical and mental capacities tend to diminish, and there is an increase in the time spent in the closest physical environment. Parks, green areas and natural environments, as well as direct contact with nature, are beneficial since they favour the performance of daily life activities and the development of social relationships [39,40]. Thus, the natural environment becomes an extension of the home, creating emotional ties with the environment through experiences and memories. These images in the natural environment are a determining factor in the active aging process [38]. In contrast, the replacement of green areas by urban ones decreases outdoor leisure activities and social relationships and, therefore, has a negative impact on active aging [41]. For this reason, some studies propose social intervention programs for environmental conservation focused on older people [42] and, recently, there have also been some studies that evaluate the few programs aimed at older persons and the environment [43].

It should be added that, despite their importance for older people well-being and quality of life, studies that evaluate the environment through images, whether digital or not, are scarce. Therefore, it is pertinent for research to focus on the way aging is experienced while acknowledging aspects such as gender, the environment and images that reflect old age [6], as well as considering population aging as an enrichment strategy for societies [43] and broadening the scope to environmental gerontology, the geography of aging, physical activity in the natural environment, age-friendly cities, etc., to mention a few of the consulted studies.

## 2. Materials and Methods

Even if we deal with percentages and figures, a qualitative methodology has been carried out using virtual ethnography, also known as digital ethnography or netnography, as a research technique, which can be defined as a "qualitative research method that adapts ethnographic research techniques to study cultures and communities that emerge through computer-mediated communities" [44]. Through this qualitative digital research technique, whose pioneer is Christine Hine with her work Virtual Ethnography (2000) and in Spain its main reference is Elisenda Ardevol [45–47], a descriptive knowledge of the cultural artifacts [48,49] of interest is obtained; in this case, from digital images retrieved from open access image banks.

An image bank is an online repository and important audio-visual documentation tool [16] of photographs, illustrations and other visual materials that can be accessed, viewed and downloaded for use in a variety of settings. Bearing in mind that there is a wide range of repositories in the virtual space, a series of selection criteria have been established for image banks, which must be: 1. open access, i.e., they must allow images to be viewed and downloaded, and 2. they must have royalty-free images (creative commons) or have the possibility of downloading them for non-commercial use. However, in some cases access is restricted and this may be due to having to register on the platform, attributing authorship by means of a link or subscribing, which implies a financial outlay. Taking these limitations into account, the chosen image banks were Freepik, Storyblocks, Canva, Pixabay and Pexels.

The list of selected concepts, in Spanish and in English, came from more extensive lists. These initial broader glossaries were based on previous studies, both by the authors (their projects, theses, papers) and from other publications and experts on these topics. The final list was also used to substitute words when a concept in an image bank did not show any results. In this way, the following terms were searched in Spanish: "personas mayores" (n = 35), "vejez" (n = 27), "edadismo" (n = 17), "tercera edad" (n = 14) and "anciano/a" (n = 3) and its correlate in English: "elderly" (n = 18), "ageism" (n = 10), "retired people" (n = 26). Other concepts included in the above-mentioned list are: "retired", "elder", "old age", "older people", "older persons", "ageing" and "aging". A few more have recently been incorporated and are being evaluated in a second phase of this research, currently in progress, to broaden the sample. It should be noted that during the virtual fieldwork, the most frequent search suggestions made by people who consult these platforms, which are linked to the introduced concept, were also recorded. In other words, the alternative suggested by the algorithm was included. For example, when searching for "elder people" the virtual image bank recommends searching for "elder people sunset", "older/elder people care", "older/elder people activities", among others. This "spontaneous" pattern offered by the platforms reveals how users search for images that represent older people and, furthermore, it can be considered as an indicator of expressions that are associated with older persons.

The following criteria were defined for the search and selection of images: 1. They must be photographs of people, i.e., other audio-visual materials such as illustrations or videos were not selected, and 2. The filters "most popular" or "most downloaded" were applied in each search, depending on the terminology and options of each image bank.

They did so in order to evaluate the most viewed or downloaded images and were therefore more likely to have been used in personal, academic or work projects, among others.

Once the search has been performed, the selected images are the first ones to have met the above criteria. To avoid any bias linked to the subjectivity of the researchers, the number of images selected was previously determined randomly by an automatic generator. As is often the case in international image banks, some searches do not give back any results, in which case the term is replaced by another from the available list.

The fieldwork was conducted between December 2021 and February 2022. The sample obtained was made up of 150 images of older people that had been extracted from the previously indicated image banks. These audio-visual materials, which reflect situated views [48] on aging, have been stored in our own database as a way to solve the problem of the ephemerality of data in digital environments. In addition to archiving the images, the characteristics observed in the images themselves and in the metadata, together with the link for their identification, have been recorded.

With the contrasted design, each of the selected images was evaluated individually following the structure of 22 variables that describe the people who appear in them, and the environment and the activities they are carrying out (Table 1). Likewise, the following metadata have been recorded to collect additional information to contextualize the content reflected in the images (keywords and title of the images) and to help locate them in the image bank (suggestion box and filter applied, in this case, the most downloaded).

**Table 1.** Variables used for image evaluation (n = 150 images).

| Sex | Type of Group | Skin Colour and Other Physical Traits | Teeth | Hair | Skin | Clothing | Dependency | On Whom It Rests | Disability and/or Use of Devices/Prostheses | Emotional State |
|---|---|---|---|---|---|---|---|---|---|---|
| Woman | Intergenerational | White | Not visible | Bald | Wrinkled | Not visible | Dependent | Older person | Not visible | Unrecognizable |
| Man | Intragenerational | Black | In good condition | Long | Stretched | Sport | Independent | Grandson/ Granddaughter | Wheelchair | Anger |
| Nonbinary Man and woman | Does not apply | Asian | In bad condition | Dyed | With make-up on | Casual/informal | Unknown | Other | Cane | Joy |
| Two men | | Oriental | Denture | Short | Aesthetic retouching | Formal | | | Jaw protheses | Surprise |
| Two women Group of women Group of men Mixed group | | Albino Vitiligo Interracial Mestizo Indigenous B/W picture | | White Grey Transition Not visible | Not visible | Mourning attire | | | Audiphones Glasses Electric lift Blind Leg prothesis Arm prothesis Bedridden person | Disgust Sadness Fear Unrecognizable: facemask |

| Image planes | Foreground | Objects | Relational gesture | Natural space | Space | Space description | Activity | Performed activity | Activity description | Literal title of the image in the source |
|---|---|---|---|---|---|---|---|---|---|---|
| One person | The older woman | Technological | Physical contact | Yes | Residence, residential centres | Open response | Passive | Cultural | Open response | Open response |
| Several people on the same plane Several people, but one stands out | The older man | Sanitary | Without contact | No | Housing/house | | Active | Sport | | |
| | The young person | Sport | | Unknown | Collaborative housing | | Portrait | Technological | | |
| | | Decorations (flowers) Household items Without objects Others: open response | | | Urbanised street | | Other: open response | Domestic | | |
| | | | | | Beach Park | | | Care Communicative | | |
| | | | | | Garden | | | Portrait | | |
| | | | | | Desert Sea | | | Leisure Work Passive action (sitting down) | | |
| | | | | | Mountain | | | | | |
| | | | | | Countryside Forest No background Other: open response | | | | | |

## 3. Results

As a result of the fieldwork, a total of 150 images have been evaluated. However, given that in many cases the images represent couples or mixed groups, more than 200 older people have actually been observed. In addition, it should be clarified that this is a qualitative study that aims to observe, for example, various aspects of daily life and the territorial location where older people are represented have been addressed Thus, the results obtained make it possible to shape and interpret the image of old age that is created and transmitted in different areas, based on the use of images from open access and free downloadable image banks.

### 3.1. Older People in Digital Images: General Features

Firstly, and in response to the question "How are the older people represented in digital images?", physical and emotional characteristics have been evaluated based on the variables observed in the images themselves and/or in the metadata.

In the 150 images evaluated, older persons appear in the images accompanied by other older people (see Figure 1), family members or health professionals (n = 86). In these cases, the representation of couples composed of a man and a woman of the same age (n = 44), that is, intragenerational couples, stands out. On the other hand, in 64 of the images older people appear alone. More precisely, older women are frequently shown represented alone (n = 35) or accompanied by other women of different ages (n = 16), and older men appear in the images mostly alone (n = 27) or together with other older men (n = 11). As a result of the above, it is worth looking more closely at who is accompanying the older persons. It has been observed that the images show older people together with persons of the same age (n = 55) or they appear in the company of younger people (n = 31).

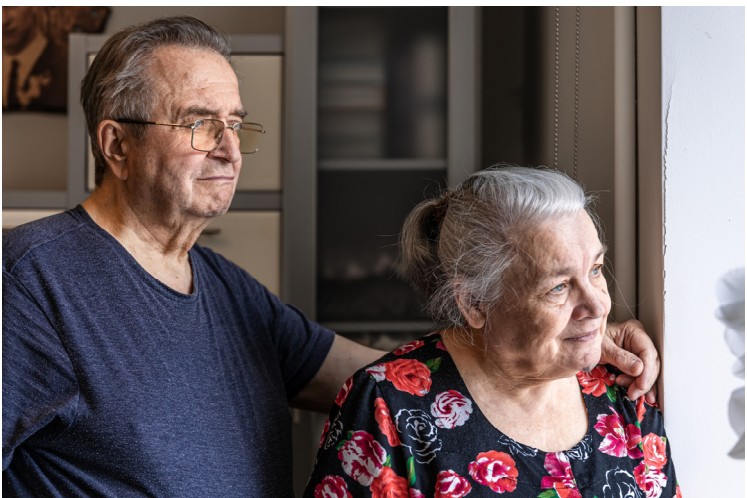

**Figure 1.** Couple of older persons. Source, Freepik. https://www.freepik.es/foto-gratis/linda-pareja-ancianos-ventana-busca-alguien-esperando_15203032.htm (accessed on 22 November 2022).

Another aspect of interest that has been noticed in the images is the plane in which the older persons are placed. The plane is understood as the position that a person occupies in the photograph with respect to other people. A person can be found in the foreground, making their presence stand out, or in the background, behind other people. It has been noted that in the case of couples or groups, they are found in the same plane or have greater prominence, either because the focus is on one person or because they are doing something while the others are looking at them. In 61 of the cases, the people who appear in the images are on the same plane while in 25 of them only one person stands out. In the latter, this tends to occur more frequently in couples consisting of a man and a woman, with the older woman being in the foreground.

In terms of physical appearance, out of the approximately 231 older persons who appear in the images evaluated, 181 are white and the rest are distributed among people of African descent (n = 14), Asians (n = 14), Middle Easterners (n = 6) and Latin Americans (n = 2). Out of the remaining 14, 6 are black and white images and could not be identified and 8 images show people of different ethnicities together.

As seen in this image, in 134 of the cases people's teeth are not visible and when they are, 89 are in good condition (3 appear to be false teeth) and only 5 are in poor condition. Another characteristic observed is their hair, with similar results: 76 of the older persons have white hair, followed by people whose hair is in the process of turning grey, which has been called "grey transition" (n = 71), 7 are bald, 8 of the older people have short hair, 12 have long hair and 19 have dyed hair, particularly women. Thirty-eight of the cases could not be identified as any kind due to their hair not being visible.

As with teeth, another aspect that can be observed in the images is the appearance of the skin of the older persons. The results show that 120 have wrinkled skin, 23.8% have stretched skin, 15 have make up on (it should be noted that sometimes these are photographic portraits taken in studios, so that people's appearance is often touched up) and in 9 of the images there seems to be some aesthetic retouching. In the remaining 32 images, the skin is not visible or does not allow a clear classification.

Finally, the clothing of the people appearing in the evaluated images is mostly casual/informal (n = 174), followed by formal (n = 28), not visible (n = 20) (for example, portraits of hands), eight sportswear and only one are mourning clothes.

### 3.2. Older People: Dependence and Care

When evaluating the images, the focus was also put on the activities the seniors were doing. Sixty-one are photographic portraits, i.e., they are stock photographs created in a studio. Even so, in 51 of the images the older people are performing some active action or activity that requires physical movement (see Figure 2), compared to the 32 that are reflected without performing any activity. The remaining 6 are images that focus on the hands of older people and since there is no significant metadata available, it is not possible to make an accurate classification.

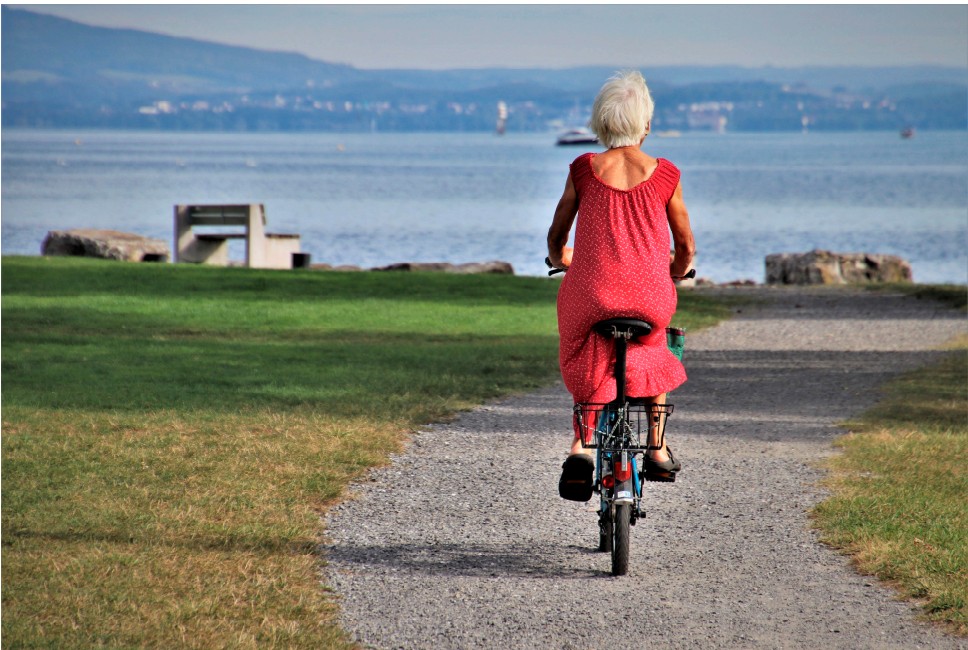

**Figure 2.** Older woman cycling. Source, Canva. https://pixabay.com/es/photos/parque-la-bicicleta-mayor-solitario-5528190/ (accessed on 20 November 2022).

Based on this distinction, different kinds of activities can be identified and it can be seen more specifically that 43 of them are portraits, in 22 images they are shown to be performing leisure activities, 20 show passive actions, 16 are care activities, 11 are sports activities, 11 show activities related to the use of new technologies, 9 are communicative activities, 7 are domestic, 6 are work-related activities and finally, 5 of the images show cultural activities.

The images also show a difference in terms of the activities carried out by older people according to sex. In this sense and focusing on the cases in which the older persons are alone, regarding men, the activities shown are mainly related to, for example, new technologies (n = 4), leisure (n = 4), work (n = 3) or cultural activities (n = 2). On the other hand, older women are frequently shown to be engaged in passive action activities (n = 8), sports (n = 4), domestic (n = 3), leisure (n = 2) or care activities (n = 1). Depending on the type of group, it is observed that in the images showing a group of men, the activities they carry out are mainly leisure activities (n = 6) while women are represented carrying out care activities (n = 6). In the case of couples made up by a man and a woman, they appear performing care activities (n = 7), passive action activities such as sitting on a bench or sofa or contemplating a natural landscape (n = 7), leisure activities (n = 7) or sports activities (n = 4), such as walking in natural environments or stretching exercises.

Another aspect of interest that the evaluated images allow us to notice is the relationship between care and dependence (see Figure 3). In this sense, it can be observed that the representation of auxiliary technological devices is infrequent: of the people observed in the 150 images, 171 appear without any aid device for mobility and/or prosthesis. Although scarcely, in some images older people use glasses (n = 46), a cane (n = 10), a wheelchair (n = 3), or are in a hospital bed (n = 1). Moreover, in 26 of the 150 images, older people are being cared for by a younger person (n = 20) or another older person (n = 6). Two images show dependent but lonely elderly people and six of the images show older people caring for a minor, which according to the metadata are their grandchildren. However, 116 of all images show independent older people.

Finally, in the images evaluated it was observed whether there was physical contact and the emotional state of older people. Out of the total number of images evaluated, in 57 of them there is visible physical contact, of which in 37 images it is people from the same generation and in 20 it is between people of different ages. The most frequent relational gesture observed is linked to hands and hugs. In terms of their emotional state, older people in the images are mainly shown to be cheerful (n = 124). In 91 of the images, no emotion is shown as they have their backs turned, are wearing masks or because the photograph focuses on their hands. The rest of the emotions represented was sadness (n = 10), surprise (n = 2), anger (n = 2), and fear (n = 1).

### 3.3. Older People: Environmental Issues and Land

Another objective of this digital research was to observe the spaces in which older people are located in the images evaluated. In 44 of them it is unknown, either because they are portraits without a background or because there is no additional information to contextualize the images. Even so, 68 of the older persons are represented in outdoor spaces, and 38 in indoor spaces, mainly in housing (n = 21) or healthcare environments (n = 6) while only 2 are shown in residences.

As for outdoor spaces, older people appear mainly in urban spaces (n = 43) followed by natural/rural spaces (n = 25) (see Figure 4). Specifically, the spaces in which older people are most often located are parks (n = 22), gardens (n = 11), in the urbanized street (n = 10), in the countryside (n = 8), in the forest (n = 7), at the beach (n = 6), in places near the sea (n = 2), in the mountains (n = 1) or in the desert (n = 1).

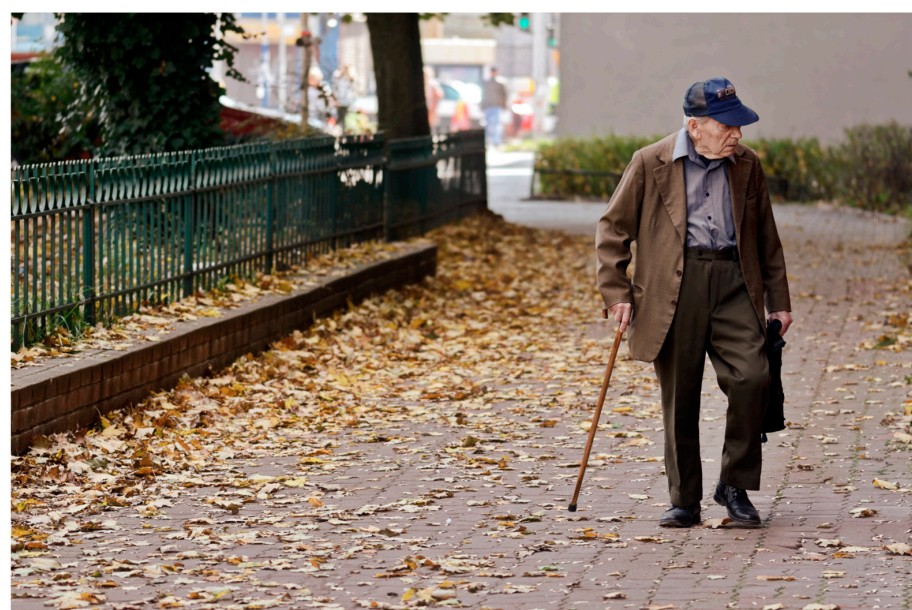

**Figure 3.** Older man with a cane. Source, Pixabay. https://www.canva.com/photos/MADyQ5rAJ-0-photo-of-elderly-man-walking-on-pavement/ (accessed on 20 November 2022).

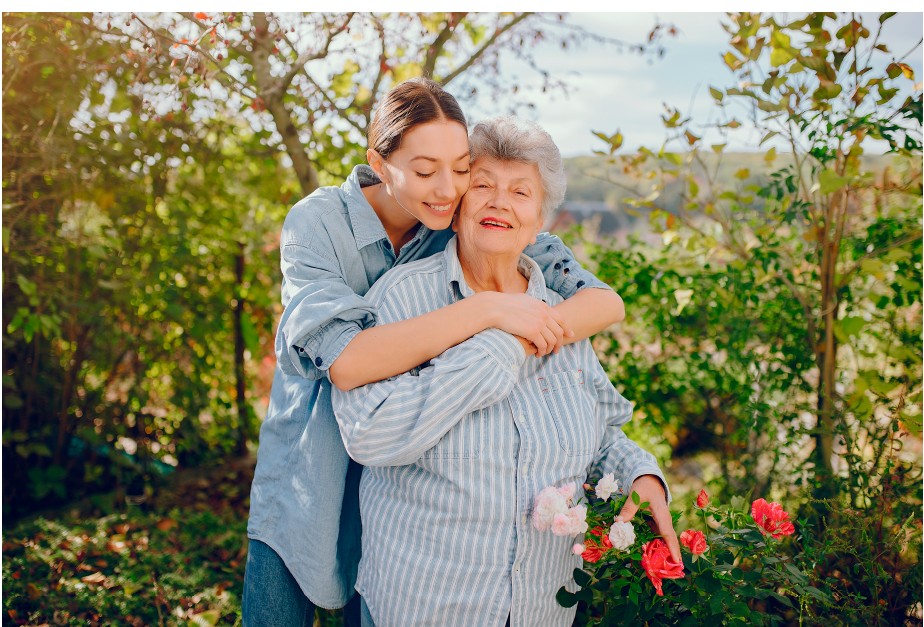

**Figure 4.** Older and younger women appear in natural space. Source, Freepik. https://www.freepik.es/foto-gratis/anciana-jardin-joven-nieta_7397387.htm (accessed on 20 November 2022).

In indoor spaces, 22 of the older people appear in couples or groups, 11 women and 5 men. In outdoor spaces, the distribution according to sex is similar, although older women are more represented in urban environments (n = 16) compared to men (n = 12), and in natural/rural environments there are more men (n = 8) than women (n = 7).

As for the activities that older people perform, in the case of natural environments they are sports (n = 6) or leisure activities (n = 3) and, in urban environments, they are mainly leisure activities (n = 15) and passive actions (n = 4).

Finally, regarding the emotional state of older people according to the space in which they are located, the images show older persons are happier outdoors (n = 60), especially in parks (n = 25). In contrast, sadness appears more in images in which the older persons are

at home (n = 4) or healthcare environment (n = 1), that is, in indoor spaces. In the other cases the space is unknown since they are portraits.

## 4. Discussion and Conclusions

This study, by means of virtual ethnography applied to a set of 150 images obtained from open-use image banks, attempts to answer how old age and aging are constructed in the collective imaginary. First of all, attention has been paid to how older people are represented in the analysed images.

It has been observed that there is a feminization of old age [25] since there are many more women portrayed than men, and they usually have more prominence in the images, as they are placed in the foreground.

Another one of the highlights has been the representation of social relationships that older people maintain. Perhaps the most surprising result has been not having found a single image in this sample of a group of women from the same generation. It could be considered as friendship groups of the same age seems to be something exclusive of male old age and not so of female old age. Older women appear alone or in the company of family members (daughters/sons, granddaughters/grandsons) or health professionals, but never in the company of other older women that we could interpret as friends. Unlike women, the men depicted in these images do have a social life, and the leisure activities they engage in are evidence of this gender stereotyped representation.

Then, the image of loneliness, usually attributed to old age and aging [32,33], is relatively absent, and, in fact, in most of the images older people appear accompanied. This representation of older people is predominantly accompanied by an old age idealization that carries certain risks as it may produce frustration in older persons when their experience differs from this representation [34].

The company and the activities they carry out determine the aging of the person, since social and family relationships are key aspects for the promotion of leisure and free time activities that help promote active aging [38]. Therefore, a highly differentiated representation of these aspects between men and women has very relevant implications, both for self-image and for older people's identity as a better valued group, without biased images that are not excluded by gender, age, space, or other key variables [1].

The analysed images portray people who mostly perform active activities, so it can be stated that they show a positive image of aging, an active aging. Leisure and care are the most represented kinds of activities, as well as passive actions which are mainly portraits. However, differences are found between the main activities performed by men and women. The little representation shown in the images of the labour market is starred by men, as is the use of technology or leisure. Women, on the other hand, are portrayed performing mainly passive, sporting, and domestic activities. In addition, they are represented carrying out care activities, but men are hardly seen performing this kind of activity. Moreover, older people are observed doing sports activities (such as cycling, walking, stretching . . . ) in natural spaces, which again demonstrates the representation of active aging. It is mainly women who carry out this type of activity, which is a solitary leisure activity. Leisure is also very much determined by the space in which it takes place.

In terms of dependency and care, older people are shown in these images are mostly independent. This is reflected by the infrequent representation of assistive technological devices, although they actually have a very positive impact on the independence of older persons [50], and why older persons are depicted as caregivers but also as recipients of care; they care for and are cared for. In addition to this, the images illustrate active aging in the activities captured, as mentioned above. It is important to highlight the representation of hands in these images, which seems to allude to care, as they are hands that support other hands.

In the analysed sample, a positive representation of old age as a non-decadent vital stage has been observed, showing cheerful older people, accompanied by others, with physical contact, and carrying out activities. It is an image close to the objective of active

aging [33,34], but also a stereotyped one, as indicated by different studies [3,5], allowing us to glimpse at prejudices associated with this group. In addition, since most of the images are stock photographs, that is, photographic portraits, older people are portrayed with a neat appearance, ranging from their clothing to their physical features (hair, teeth...), which reflects the good health of these people.

Also, most of the people who appear in the images evaluated are white, well-groomed and in couples. It can also be interpreted that, in the cases in which there is also physical contact and/or some object such as a ring, they could be heterosexual couples living together in the same household. With this, it can be concluded that the representation of old age continues to be unvaried [6], as it excludes homosexual or interracial couples, non-binary, and racialized people; rendering invisible numerous minorities in old age and thus corroborating that sexism, ageism and racism continue to be the main forms of discrimination [6,12]. A fundamental question in this digital research is to observe in which spaces older persons are located, and they are more frequently represented in outdoor spaces, but in urban contexts. That is, they are mainly located in parks and gardens, natural spaces transformed and adapted by and for human beings. These spaces, as well as the direct contact with nature, present benefits for older people by favouring the performance of daily life activities and the development of social relationships [39,40]. This is reflected in the images as they show that the leisure activities realised in these spaces consist of board games (dominoes, chess, or cards), dancing, playing with grandchildren, or sightseeing.

On the other hand, older women are mostly represented at home while men are mainly depicted in natural/rural spaces, a phenomenon that is probably due both to the fact that they are doubly stereotyped for being women and for being older [29], and to the masculinization of old age in rural areas [25].In general, there is little representation of older people in residences, and this limitation should be highlighted because it is difficult to distinguish homes from residences.

In the evaluated images older people are not directly shown caring for the environment; it is an absent representation: Older persons are portrayed in natural urban spaces. These images reflect a certain proximity to the natural environment. Nonetheless, older people represented as active actors involved in the care and conservation of the environment is practically non-existent, so it would be promising to promote programs with these objectives focused on older people as proposed by some studies [43].

Finally, the emotional state reflected, in these images, is fundamentally one of great joy, which indicates a positive perspective on old age [51] and shows us, in short, a representation of the well-being and quality of life of this group [35,37]; even though it may also reflect a stereotypical view of old age [34].

It is also noteworthy that in natural/rural environments, older people appear happier than in urban spaces. As for the research itself, it should be added that a second phase of fieldwork is in progress to check whether or not the same trends are being followed, update data and contrast information with a larger sample of images. In addition, the field of study has also been diversified to include other image banks such as Gratisography, Pinterest, Unsplash, SplitShire or OpenPhoto.

As for the specific objectives and design aspects, similar sampling, recording and storage decisions have been made as the ones applied in this first phase of the digital social research. Virtual ethnography also poses specific challenges, such as the problems of access and use of certain cultural artifacts [48], making it difficult to evaluate them even if they are of interest to the study. Another methodological limitation is the problem of data ephemerality, and, for this reason, archival storage decisions have focused on archiving the evaluated images in addition to recording the direct access link. Other study limitations are the need to: (1) analyse other image banks to contrast the representations of older people at an international level, (2) increase the sample of images to be able to, in the future, speak of a greater numerical representativeness, and (3) the trouble that results from the difficulty of different evaluators applying the 22 variables or labels (e.g., about their skin) and agreeing on them. This last methodological limitation was overcome over the course

of various debates and reflections that led to the reformulation of some categories and the re-evaluation of some images. Despite all this, the advantages of the study are based on the accessibility and richness of the data and on the fact that it is an ethically non-invasive social investigation.

**Author Contributions:** Conceptualization, G.L.C., M.S.A.-T. (original idea and coordination of meetings) and G.F.-M.; Methodology, G.L.C., S.C.-B., M.S.A.-T., V.Z.-M. and G.F.-M.; Investigation and process, G.L.C., S.C.-B., A.A. and G.F.-M.; Formal analysis, G.L.C., S.C.-B., R.H.-S. and C.M.-M.; Writing—original draft preparation, G.L.C., S.C.-B., A.A., R.H.-S. and C.M.-M.; Writing—review and editing, G.L.C., S.C.-B., M.S.A.-T., R.H.-S., C.M.-M., V.Z.-M. and G.F.-M.; Visualization, G.L.C., S.C.-B. and V.Z.-M.; Project administration, G.F.-M. and M.S.A.-T.; Funding acquisition, G.F.-M. and M.S.A.-T. All authors have read and agreed to the published version of the manuscript.

**Funding:** Ministry of Science, Universities, and Innovation, corresponding to the Program of R&D Activities between research groups of the Madrid Region, and co-financed by the European Social Fund. R&D Activities Program ENCAGEn-CM: "Active Ageing, Quality of Life and Gender. Promoting a positive image of old age and ageing, against the ageism", https://encage-cm.csic.es/ (accessed on 20 November 2022) (Ref. H2019/HUM-5698) main research G.F.M, 2020-2023 and Prof. Dr. Agulló-Tomás who has contributed with a personal provision and provisions proceeding of UC3M. This project was granted in 2022 (see more information: http://iegd.csic.es/es/article/programa-actividades-id-encagen-cm-coordinado-csic-recibe-premio-fundacion-pilares) (accessed on 20 November 2022).

**Data Availability Statement:** The following documents that have been prepared for this work can be consulted: (1) Own image bank that includes the 150 evaluated images https://drive.google.com/drive/folders/1HAYo4Shx_6N-qnk_CRR3OwCJXW07yC1d?usp=sharing (accessed on 20 November 2022), (2) List of some diverse studies on ageing and gender https://docs.google.com/spreadsheets/d/1lkyun1NaM3xH-dqAHIzqmpCtjmtc2m66fAV4w58Nb5Y/edit?usp=sharing (accessed on 20 November 2022) and (3) List of some references on older people and the environment https://docs.google.com/document/d/1fdcWW5-fFcMqpI2qA9RbfslzGSUV5Mv64mQ_wrp3mk0/edit?usp=sharing (accessed on 20 November 2022).

**Acknowledgments:** The authors appreciate all those who contributed to this work for their support and dedication: to María Loro Rubia for her support during the fieldwork and to Mercedes Sánchez Millán for the translation of the article.

**Conflicts of Interest:** The authors declare no conflict of interest.

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
