# Peer review of "Evaluation of Older People Digital Images: Representations from a Land, Gender and Anti-ageist Perspective"

_land, doi:10.3390/land12010018_

Round 1

Reviewer 1 Report

I think this is a good article and an important topic. I very much like the work the researchers have done - it is a new area of research and ageism is so important to address. I believe with some small editorial and additional information this should be published.

Comments: the authors should state the purpose more clearly - they discuss women, intersectionality, and environment but do not really discuss these in depth - the environmental relationship in particular could be addressed more fully with research in the area mentioned. The tables are in Spanish and need translation it was difficult for me to understand what the findings were given that I am not fluent in Spanish.

Author Response

Thank you very much for your comments and suggestions to all reviewers. We will reply to each of them in blue below and the changes have also been noted in the paper itself. Thank you again and best regards,

The authors

Reviewer 1

I think this is a good article and an important topic. I very much like the work the researchers have done - it is a new area of research and ageism is so important to address. I believe with some small editorial and additional information this should be published.

Comments: the authors should state the purpose more clearly - they discuss women, intersectionality, and environment but do not really discuss these in depth - the environmental relationship in particular could be addressed more fully with research in the area mentioned. The tables are in Spanish and need translation it was difficult for me to understand what the findings were given that I am not fluent in Spanish.

Thank you very much for your positive feedback and recommendations. The missing words/tables have been translated into English. We apologise for the inconvenience. Thank you again for your interest and evaluation which clearly supports the publication of our work.

Reviewer 2 Report

Review of “Environmental Issues, Land, and Older People. Images’ evaluation from a gender, anti-ageism, and intersectional prism.”

Thank you for the opportunity to review this manuscript. I do find this topic and methodology to be intriguing and potentially worthwhile for publication. I do have several comments (often with questions for the authors to address) that would strengthen the manuscript. I do need to disclose that I am unable to review the references page or any of the tables because they were not in English.

Here are my suggested comments/edits:

1 .Background section. This section is well-written and includes some excellent points. I do suggest a bit of re-organization and additional information, potentially with headings, so that the authors can make it very clear why (and how) the focus of the image analysis is on gender, anti-ageism, and an intersectional prism (as stated in the title). Because the abstract also mentions analyzing the images from a socio-spatial and land perspective, I also suggest that the background section clearly establishes why this is. Then, moving this forward, in the Results section, the authors need to be very clear about how they analyzed the images from a gender, anti-aging, intersectional, socio-spatial, and land perspective.

2. Materials & Methods section, page 4. Can the authors explain why they only present percentages for the findings? If using qualitative methodology, this seems very problematic to mostly focus on presenting percentages to explain the findings. And if you do only present %s as the article currently does (meaning you don’t make any changes to how the findings are presented), make sure to cite a resource justifying this approach within the first paragraph of the Materials & Methods section.

3. Materials & Methods section, page 4. What was the training & backgrounds of the image evaluators? How, for example, would one know the difference between wrinkled skin and stretched skin? Did the evaluator go through training, or perhaps they had a dermatology background? Also, did the evaluators examine agreement across evaluators for the various determinations?  For example, did the evaluators ensure 80% agreement when reviewing the images, or was there another technique or statistic used?

4. Materials & Methods section, page 4. How were the image banks selected? Are these the mostly commonly used ones by media outlets, for example? Who has access to the ones chosen? If these are paid sites, the authors need explain who would pay for access to these sites & why this is important to analyze. In my mind, it would be more advantageous to utilize image banks that are freely available to the public in order to better capture what images people see when they, for example, google “Older people.”

5. Materials & Methods section, page 5. How did the authors find the images? Did the authors add new images once every couple of weeks between December & February of 2021 through 2022, or were the images extracted at different points in time over this time period? How many people from the research team were involved with this?

6. Results, page 6. There is a sentence that reads, “In addition, it should be clarified that this is a qualitative study that aims to observe, for example, the appearance of the older people, the spaces in which they are placed, in whose company they are, which activities they are performing, as well as their mood, the presence of technological or sanitary devices, among other things.” This needs to be stated much earlier in the article. This sentence seems to explain what the focus of this study is about & the type of information that is being analyzed, but this is not well-stated in the article until this point.

7. Results, page 7. The authors state that the image is in the “plane” in which older persons are placed. What is meant by “plane”? Make sure to include a definition for readers, like myself, who are less familiar with this term.

8. Discussion, page 10. Avoid one-sentence paragraphs. The authors should consider re-organizing this whole page of the discussion. It seems a bit disjointed. Perhaps some headings would help point out the main themes?

9. Discussion, page 11. Going back comment #4, the Discussion could be a great place to further include information & limitations about the image banks selected. I am wondering about Google Images, for example. Any reason why that one wasn’t selected? Just make it very clear to readers which ones you selected and why, and what some of their benefits & limitations are.

Author Response

Thank you very much for your comments and suggestions to all reviewers. We will reply to each of them in blue below and the changes have also been noted in the paper itself. Thank you again and best regards,

The authors

Reviewer 2

Thank you for the opportunity to review this manuscript. I do find this topic and methodology to be intriguing and potentially worthwhile for publication. I do have several comments (often with questions for the authors to address) that would strengthen the manuscript. I do need to disclose that I am unable to review the references page or any of the tables because they were not in English.

Thank you very much for your very positive feedback and recommendations. The missing words/tables have been translated into English. We apologise for the inconvenience. Thank you again for your interest and evaluation which clearly supports the publication of our work.

Here are my suggested comments/edits:

  1. Background section. This section is well-written and includes some excellent points. I do suggest a bit of re-organization and additional information, potentially with headings, so that the authors can make it very clear why (and how) the focus of the image analysis is on gender, anti-ageism, and an intersectional prism (as stated in the title). Because the abstract also mentions analyzing the images from a socio-spatial and land perspective, I also suggest that the background section clearly establishes why this is. Then, moving this forward, in the Results section, the authors need to be very clear about how they analyzed the images from a gender, anti-aging, intersectional, socio-spatial, and land perspective.

Thank you, we have included in the abstract the other approaches used in addition to the socio-spatial one and connected them all in the results. In the selection of criteria/variables and bibliographical references that have guided all this, the use of these approaches in the paper and previous documents that support it, both by the authors and other referenced authors, is noted. 

  1. Materials & Methods section, page 4. Can the authors explain why they only present percentages for the findings? If using qualitative methodology, this seems very problematic to mostly focus on presenting percentages to explain the findings. And if you do only present %s as the article currently does (meaning you don’t make any changes to how the findings are presented), make sure to cite a resource justifying this approach within the first paragraph of the Materials & Methods section.

Thank you for your comment. We have considered it appropriate to indicate the results in percentages because the volume of images evaluated is high and also because the system for recording and storing the data allowed for more detailed descriptions. However, we believe that your assessment is correct and for this reason, we have updated all the data in the results section.

  1. Materials & Methods section, page 4. What was the training & backgrounds of the image evaluators? How, for example, would one know the difference between wrinkled skin and stretched skin? Did the evaluator go through training, or perhaps they had a dermatology background? Also, did the evaluators examine agreement across evaluators for the various determinations?  For example, did the evaluators ensure 80% agreement when reviewing the images, or was there another technique or statistic used?

Yes, the lead evaluators who have carried out the fieldwork (evaluation of the 150 images) have followed the criteria/variables agreed by the whole evaluation team. They all have a sociological background (degree in Sociology) and have worked in social research techniques and programme evaluation. After several meetings and "review of judges" (we have agreed among ourselves and consulted with other references/colleagues) to test the different "variables", they have been adapted and selected to make a better consensual and joint evaluation. For example, to differentiate between artificially stretched skin (with aesthetic retouching) and naturally wrinkled skin has been defined in the article. It has been found that 51.9% have wrinkled skin, 23.8% have stretched skin.

  1. Materials & Methods section, page 4. How were the image banks selected? Are these the mostly commonly used ones by media outlets, for example? Who has access to the ones chosen? If these are paid sites, the authors need explain who would pay for access to these sites & why this is important to analyze. In my mind, it would be more advantageous to utilize image banks that are freely available to the public in order to better capture what images people see when they, for example, google “Older people.”

Yes, the images are freely accessible, and we agree with you that it is more appropriate to use this criterion in all respects. We have added the names of the image bases in the abstract as well.

  1. Materials & Methods section, page 5. How did the authors find the images? Did the authors add new images once every couple of weeks between December & February of 2021 through 2022, or were the images extracted at different points in time over this time period? How many people from the research team were involved with this?

The 150 images were selected between December and February 2021-2022, as modified on page 5, in the penultimate paragraph.

Three components of the evaluation team carried out this fieldwork, but the whole team participated in the meetings in process because some doubts arose about how to select or when to point out such criteria, and it was agreed and decided by the whole group, meetings coordinated by the PI of the UC3M research team (see the part of distributed tasks, this has been added in the same).

  1. Results, page 6. There is a sentence that reads, “In addition, it should be clarified that this is a qualitative study that aims to observe, for example, the appearance of the older people, the spaces in which they are placed, in whose company they are, which activities they are performing, as well as their mood, the presence of technological or sanitary devices, among other things.” This needs to be stated much earlier in the article. This sentence seems to explain what the focus of this study is about & the type of information that is being analyzed, but this is not well-stated in the article until this point.

Yes, this sentence has also been incorporated before, in the Method section.

  1. Results, page 7. The authors state that the image is in the “plane” in which older persons are placed. What is meant by “plane”? Make sure to include a definition for readers, like myself, who are less familiar with this term.

A brief clarification on this concept has been incorporated in the text. We hope that what we intended to explain is now clearer.

  1. Discussion, page 10. Avoid one-sentence paragraphs. The authors should consider re-organizing this whole page of the discussion. It seems a bit disjointed. Perhaps some headings would help point out the main themes?

Efforts have been made to reorganise it in a more coherent way.

  1. Discussion, page 11. Going back comment #4, the Discussion could be a great place to further include information & limitations about the image banks selected. I am wondering about Google Images, for example. Any reason why that one wasn’t selected? Just make it very clear to readers which ones you selected and why, and what some of their benefits & limitations are.

Yes, it has been made clearer which images have been selected and which have not. E.g., Google images have not been evaluated because they do not form an image bank as such (it is specified in the paper which ones have been analysed), but Google has broader functionality, unlike image banks themselves, whose main function is to group and offer images for different uses. Perhaps some of the images in the image banks studied coincide with those that can be searched for in Google. In any case, we take good note of the recommendation and, as we are in a second phase of the fieldwork, it may be worth considering it for future work. Thank you again for your recommendations.

Reviewer 3 Report

This paper addresses the relationship between images and evaluation from intersectional gender and anti-age perspectives. For which it applies a virtual ethnographic methodology. It is based on the analysis of repositories of images on the Internet, which are cross-checked and classified for their interpretation in terms of content. 

The main problem with this article is that it is not in the focus of the objectives and themes that are worked on by Land Journal. The authors do not make an interdisciplinary effort to effectively understand the relationship between images and their socio-spatial dimension. They continue to understand the spatial only as a backdrop in visual expressions.

While it states that it seeks to understand the environmental and terrain issues that are linked to the image. The article fails to acknowledge emerging fields of spatial discussion of images. For example, the field of visual geographies, developed by authors such as Gillian Rose, on audiovisual methods. Nor does it use categories proper to visuality in spatial studies, such as the notion of landscape, addressed in traditional works such as those developed by Denis Crosgrove. Nor does it address the notion of environmental digital humanities, which works with digital image resources to understand the challenges of generalized environmental change.

The methodology is weak and unattractive, based solely on secondary internet sources. There is no methodological novelty. Nor is there an effort to go beyond the limitations presented by the content analysis of the images, which can be carried out from the perspective of discourse analysis or critical discourse analysis. This contradicts the ethnographic posture that states that a phenomenological perspective should investigate the social meanings that people construct about these images from their experiences.

The structuring of the article and presentation of topics is also inappropriate. For example:

- The title is too long, and becomes confusing, it should be simplified and redirect the focus. 

- The introduction does not make clear what the problem to be studied is. It only focuses on describing the relevance of image analysis in the social sciences. 

- There are many ideas in parentheses that need to be integrated into the text and explained in detail. 

- In the background section, there are several abrupt changes of topic. Subheadings are needed. 

- The objectives are stated in the results section. 

- The background describes aspects that should be in a characterization of the case study.

- It should develop, cite and explain in detail what is meant by: "environmental gerontology, the geography of aging, physical activity in the natural environment, age-friendly cities, etc."

- The tables are in Spanish. They do not have a title. They are not appropriate.

- The objective is presented late, only in the results section it is stated what the authors are asking themselves. 

In summary, it is recommended to reject the article. I believe that the authors can carry out a substantial improvement of the document and send it to another journal. 

Author Response

Thank you very much for your comments and suggestions to all reviewers. We will reply to each of them in blue below (see the enclosed document) and the changes have also been noted in the paper itself. Thank you again and best regards,

The authors

Reviewer 3

This paper addresses the relationship between images and evaluation from intersectional gender and anti-age perspectives. For which it applies a virtual ethnographic methodology. It is based on the analysis of repositories of images on the Internet, which are cross-checked and classified for their interpretation in terms of content. 

The main problem with this article is that it is not in the focus of the objectives and themes that are worked on by Land Journal. The authors do not make an interdisciplinary effort to effectively understand the relationship between images and their socio-spatial dimension. They continue to understand the spatial only as a backdrop in visual expressions.

Thank you and we take note of your suggestions. We are sorry to hear that you have rejected the paper. We hope that there is a possibility that another reviewer can review the paper if they need to have another opinion, along with the other two positive ones.

Although I do not perceive it for the journal Land, from our point of view and considering the monograph (more interdisciplinary, lines that link Land and other areas, which was one of our objectives in coordinating this more eclectic monograph), it does fit in the journal and in the monograph especially, in line with other more open papers and current topics that have not been dealt with very much?

In fact, the other two reviewers have not objected in this sense of lack of fit in Land. In addition, the journal Land itself has invited us to take charge of a second edition of the monograph and we will also coordinate it, despite the difficulties of these processes.

While it states that it seeks to understand the environmental and terrain issues that are linked to the image. The article fails to acknowledge emerging fields of spatial discussion of images. For example, the field of visual geographies, developed by authors such as Gillian Rose, on audiovisual methods. Nor does it use categories proper to visuality in spatial studies, such as the notion of landscape, addressed in traditional works such as those developed by Denis Crosgrove. Nor does it address the notion of environmental digital humanities, which works with digital image resources to understand the challenges of generalized environmental change.

The methodology is weak and unattractive, based solely on secondary internet sources. There is no methodological novelty. Nor is there an effort to go beyond the limitations presented by the content analysis of the images, which can be carried out from the perspective of discourse analysis or critical discourse analysis. This contradicts the ethnographic posture that states that a phenomenological perspective should investigate the social meanings that people construct about these images from their experiences.

We take good note of the references you tell us about and will certainly consider them in future works (e.g., for the 2nd edition mentioned above).

We are sorry that you find this method unattractive, but it is the one used recently (already for decades, but especially in recent years with the irruption of the digital society we live in) by many authors and it is recommended to follow in order to exploit the virtual material on social issues. On "based solely on secondary internet sources", although it may seem simple, it is also an effort to select variables, to reach a consensus, to draw up a "questionnaire" to be applied to the images (as would be done with the survey or in-depth interview technique).

Secondary research is another type of research that is not only accepted but also widely recommended by various recent manuals and organisations that generate data (surveys, indicators, etc.), but researchers/evaluators need to exploit it more and better. In this case, it is not only a matter of exploiting something generated, but we have had to build an evaluation instrument specifically to evaluate images, select them with criteria from different image bases, join them together... The work has been rigorous in all senses.

The structuring of the article and presentation of topics is also inappropriate. For example:

- The title is too long, and becomes confusing, it should be simplified and redirect the focus. 

We are going to reconsider the title, but for the moment we will leave it unchanged because the other two reviewers have not mentioned anything about it and we think it is a bit long, you are right, but it gives more detail about the content of the paper.

- The introduction does not make clear what the problem to be studied is. It only focuses on describing the relevance of image analysis in the social sciences. 

Although we specified the problem from the beginning (lack of unbiased images, differences between men and women, few studies linking gender/environment/elderly....) it has been underlined and recalled again.

- There are many ideas in parentheses that need to be integrated into the text and explained in detail. 

Some of these ideas have been taken out of the parentheses but others have been left out as they are complementary or secondary information.

- In the background section, there are several abrupt changes of topic. Subheadings are needed. 

In a first version we had subtitles but there were still very short headings and we decided to integrate some of them. If required, they could be recovered. Links to the three areas covered are added so that they do not appear to be disconnected.

- The objectives are stated in the results section. 

The objectives are in the last paragraph of the introduction (p. 2, see below), in the abstract and in the results to refer to the fulfilment of the objectives. If required, they could be deleted from the results.

“The purpose of this article is to answer a series of research questions about the images associated with old age, the way in which the older persons are represented and where they are located. In answering these questions, we also consider related aspects such as dependency, the leisure and free time activities in which they participate, the social relationships they maintain, or their emotional state, along with the application of a gender, anti-ageist, intersectional and inclusive perspective.”

- The background describes aspects that should be in a characterization of the case study.

- It should develop, cite and explain in detail what is meant by: "environmental gerontology, the geography of aging, physical activity in the natural environment, age-friendly cities, etc."

We believe that these are recognised areas of work in the social sciences that do not require further definitions, which are widely used by the authors consulted. If required, some of them could be defined or the references could be expanded.

- The tables are in Spanish. They do not have a title. They are not appropriate.

They have been translated and the titles have been added to the tables.

- The objective is presented late, only in the results section it is stated what the authors are asking themselves. 

As commented in your other suggestion, the objectives are in the last paragraph of the introduction (p. 2, exposed) and in the results to allude to the fulfilment of the objectives. If required, the following could be deleted from the results.

In summary, it is recommended to reject the article. I believe that the authors can carry out a substantial improvement of the document and send it to another journal. 

As we said in your response to the first paragraphs, we are sorry that you did not see the fit in this journal and we hope that after our responses and taking into account the monograph (more interdisciplinary, lines that unite Land and other areas, which was one of our objectives in coordinating this particular monograph), it does fit in the journal and especially in the monograph, in line with other more open papers and current topics that are not treated very much...

In fact, the other two reviewers have not objected in this sense of lack of fit in Land. Thanks again for your suggestions and recommendations.

Round 2

Reviewer 2 Report

Thank you for your revisions. The manuscript is improved in many ways. I do have some remaining questions from my comments and about the manuscript-- mostly related to the discussion section.

Title, abstract & article

Make sure your focus areas are very clear throughout the article. For example, your title states that focus areas as "gender, anti-ageism, & intersectional" but then the abstracts states "gender, intersectional and socio-spatial, and land.' This all needs to be clear in the title & abstract and then carry over the narrative of the manuscript-- ideally throughout the findings & discussion section.   

Throughout the manuscript. 

The authors use the term "intragenerational" versus "intergenerational" - make sure to define intragenerational since that is less commonly used in the literature  

Reviewer 2, comment 6. 

I am not seeing in the Methods section where this sentence is included.   

Comment 8. 

The authors still include many one-sentence paragraphs. Any of the one-sentence paragraphs can be incorporated in either the preceding paragraph or the following paragraph. None of them seem necessary. 

Regarding the entire discussion section, this needs to be better organized--- ideally by your main focus areas. Make it very clear what your main findings are in each area. Your results & discussion is a bit "choppy, and should be more focused to ensure readers glean your main findings quickly and easily. If you leave the results as is (which I would suggest), then I suggest re-organizing the discussion section to make tie it in better with your main focus areas (meaning gener, activity, intersectional, anti-ageism socio-spatial, land)  

On page 8, 6th paragraph. 

Where it says "As seen in this image"-- is that the image to the left?  If so, make it very clear that is the case.

Page 9, 4th full paragraph. 

Is it clear in the images if the woman or the man is the care recipient?  

Page 12.

Need to include your study limitations.

Author Response

English language and style: (x) English language and style are fine/minor spell check required

Comments and Suggestions for Authors:

Thank you for your revisions. The manuscript is improved in many ways. I do have some remaining questions from my comments and about the manuscript-- mostly related to the discussion section.

Title, abstract & article

Make sure your focus areas are very clear throughout the article. For example, your title states that focus areas as "gender, anti-ageism, & intersectional" but then the abstracts states "gender, intersectional and socio-spatial, and land.' This all needs to be clear in the title & abstract and then carry over the narrative of the manuscript-- ideally throughout the findings & discussion section.   

Thank you for your suggestion. We have changed the title to match your comments, as it fits better with the content of the article (e.g., we have removed the word "intersectional" which, given the breadth of the multifaceted approach, we have not been able to deal with in depth). The new title will be: "Evaluation of older people digital images. Representations from a land, gender and anti-ageist perspective”.

Throughout the manuscript the authors use the term "intragenerational" versus "intergenerational" - make sure to define intragenerational since that is less commonly used in the literature.

Since it points out to us that “intragenerational” is less commonly used in the literature, we have decided to change it to the expression “of/from the same generation”. Thank you.

Reviewer 2, comment 6.

Yes, we forgot to include it, our apologies for that. We have now modified a sentence from the first paragraph of the Results section (page 8) and included this: “…various aspects of daily life and the territorial location where older people are represented have been addressed.”.

Comment 8.

The authors still include many one-sentence paragraphs. Any of the one-sentence paragraphs can be incorporated in either the preceding paragraph or the following paragraph. None of them seem necessary.

Thank you. We have joined some sentences together so that there are no paragraphs with only one sentence.

Regarding the entire discussion section, this needs to be better organized--- ideally by your main focus areas. Make it very clear what your main findings are in each area. Your results & discussion is a bit "choppy, and should be more focused to ensure readers glean your main findings quickly and easily. If you leave the results as is (which I would suggest), then I suggest re-organizing the discussion section to make tie it in better with your main focus areas (meaning gener, activity, intersectional, anti-ageism socio-spatial, land)

Thank you. We have reorganised the Discussion section according the reviewer’s suggestion. Hopefully it will be clearer and more attractive.

On page 8, 6th paragraph.

Where it says "As seen in this image"-- is that the image to the left?  If so, make it very clear that is the case.

No reference was made to the images that have been included as visual examples; they are only used as support of the ideas in the text. We have revised and reworded the sentence because it was causing confusion. Thank you.

Page 9, 4th full paragraph.

Is it clear in the images if the woman or the man is the care recipient? 

We agree with the reviewer. Based on the fact that the sample is not statistically (but it is qualitatively) representative, women appear more often than men both caring for and being cared for by other women (daughters or granddaughters).

Page 12.

Need to include your study limitations.

Yes, some were already included (e.g., data/images are ephemeral and can disappear from the platform) and they can be found in the last paragraphs of the article. We have now emphasised and better explained the need to: 1) analyse other image banks to contrast the representations of older people at the international level, 2) increase the sample of images to be able to, in the future, speak of a greater numerical representativeness, and 3) the trouble that results from the difficulty of different evaluators applying the 22 variables or labels (e.g., about their skin) and agreeing on them. This last methodological limitation was overcome over the course of various debates and reflections that led to the reformulation of some categories and the re-evaluation of some images.

Reviewer 3 Report

I thank the authors for the second round of revisions that have been made, and the clarifications on the focus of the article and its relevance to Land magazine. 

However, the authors have made only "minor changes", and the paper continues to maintain the errors that I pointed out in the previous version.  The authors have dismissed almost all of my comments, and have decided to incorporate almost none of the changes I have suggested by pointing out that "the other two peer reviewers do not agree with my assessment". It is unethical for the authors to request the removal of my evaluation because they "do not agree" with the demands I place on them as evaluators. In this regard, I must clarify that all the comments I have sent correspond to a demanding academic evaluation.

At the same time, I find it highly inappropriate from an ethical point of view for the authors to claim that my assessments can be overlooked because they have been invited to edit an upcoming issue. Precisely in their capacity as editors, their contribution should be open to receive contributions that enrich it, and not maintain a closed perspective, this is what science is all about. 

On the other hand. The problems with the article continue, particularly due to the failure to address the main point I made in my previous assessment, "The article fails to acknowledge the emerging fields of spatial discussion of the images." Pointing out that they are using a method that has been in use for decades. This is precisely the problem with this article: lack of novelty and updating with respect to international discussions. A high impact journal that is indexed in Web Of Science, should demand that authors post novel methodologies that challenge epistemological fields and at the same time be updated on the theoretical debates in their field of work. 

Finally, I would like to point out that in this new version. The authors have incorporated images taken from the Internet, which do not even indicate the source. 

Author Response

I thank the authors for the second round of revisions that have been made, and the clarifications on the focus of the article and its relevance to Land magazine.

However, the authors have made only "minor changes", and the paper continues to maintain the errors that I pointed out in the previous version.  The authors have dismissed almost all of my comments, and have decided to incorporate almost none of the changes I have suggested by pointing out that "the other two peer reviewers do not agree with my assessment". It is unethical for the authors to request the removal of my evaluation because they "do not agree" with the demands I place on them as evaluators. In this regard, I must clarify that all the comments I have sent correspond to a demanding academic evaluation.

Thank you very much for your appreciations. We have included several of your suggestions. For example, the title was too long, and becomes confusing, it should be simplified and redirect the focus. We have now modified the title: “Evaluation of older people digital images. Representations from a land, gender and anti-ageist perspective”

At the same time, I find it highly inappropriate from an ethical point of view for the authors to claim that my assessments can be overlooked because they have been invited to edit an upcoming issue. Precisely in their capacity as editors, their contribution should be open to receive contributions that enrich it, and not maintain a closed perspective, this is what science is all about.

We apologize for the confusion and thank you very much for your suggestions. Please, consider that we have attended your indications with this second revision.

On the other hand. The problems with the article continue, particularly due to the failure to address the main point I made in my previous assessment, "The article fails to acknowledge the emerging fields of spatial discussion of the images." Pointing out that they are using a method that has been in use for decades. This is precisely the problem with this article: lack of novelty and updating with respect to international discussions. A high impact journal that is indexed in Web of Science, should demand that authors post novel methodologies that challenge epistemological fields and at the same time be updated on the theoretical debates in their field of work.

We appreciate your review. Please, note that the methodology used (secondary analysis of existing material) is not new, but the approach (virtual ethnography) has recently started being applied and both the object of evaluation and type of images are new as well (virtual, platforms, which did not exist years ago). In addition, studies on images of older people from a gender perspective and considering their territorial/spatial location are scarce and therefore our article aims, with humility, to contribute to this novel union of areas that have not been treated together very often.

Finally, I would like to point out that in this new version. The authors have incorporated images taken from the Internet, which do not even indicate the source.

Thank you for your comment. The images are now correctly cited at the bottom of each photo. Therefore, the "Supplementary Materials" section (where the sources of each image were previously specified) has been removed and we now find it to be clearer.
